# Assessing the Impact of the Methane Inhibitors 3-Nitrooxypropanol (3-NOP) and Canola Oil on the Rumen Anaerobic Fungi

**DOI:** 10.3390/ani15091230

**Published:** 2025-04-27

**Authors:** Eóin O’Hara, Nora Chomistek, Stephanie A. Terry, Karen A. Beauchemin, Robert J. Gruninger

**Affiliations:** Agriculture and Agri-Food Canada, Lethbridge Research and Development Centre, Lethbridge, AB T1J 4B1, Canada; eoin.ohara@agr.gc.ca (E.O.); stephanie.terry@agr.gc.ca (S.A.T.); karen.beauchemin@agr.gc.ca (K.A.B.)

**Keywords:** rumen, anaerobic fungi, methane, cattle, 3NOP, canola oil, inhibitors, feed additive

## Abstract

Reducing enteric methane emissions from cattle is an important step toward more environmentally friendly farming. Building on our previous studies, the impact of two feed additives known to reduce methane—3-nitrooxypropanol (3-NOP) and canola oil—on an important group of rumen microbes, the anaerobic fungi. These fungi help break down fibrous plant material, aiding in digestion. Our work shows that 3-NOP reduced methane production by approximately 32% without impacting the fungal community, suggesting it is an effective option for reducing greenhouse gas emissions without affecting feed digestion. While canola oil also significantly reduced methane production, it disrupted the fungal population, causing a decline in key species that digest fiber. This disruption may explain why previous research studies have shown reduced feed digestion in cattle fed canola oil. Importantly, our study also shows that some fungal species don’t fully recover when canola oil is removed from the diet, suggesting long-term impacts on the microbiome. These findings highlight that while 3-NOP offers a targeted methane-reducing solution, care must be taken with oil-based additives. Understanding how different feed strategies affect the rumen’s microbes can help design better diets that lower emissions without sacrificing animal health or performance.

## 1. Introduction

The rumen is a large fermentation chamber located in the foregut of ruminants, harboring a complex microbiome of bacteria, archaea, protozoa, fungi, and phage [1]. These microbes interact synergistically with one another and the host animal to degrade the recalcitrant lignocellulosic biomass present in plant cell walls [2]. Microbial fermentation in the rumen meets ~70% of the host’s energy requirements, principally via the production of volatile fatty acids (VFA) [3], and provides protein and other nutrients necessary for animal performance. However, methane (CH_4_) gas produced during normal rumen fermentation reduces dietary energy availability to the animal by up to 12% [3]. Additionally, CH_4_ is a potent greenhouse gas that contributes ~4–6% of total global anthropogenic carbon emissions annually [4]. CH_4_ has a global warming potential approximately 28 times that of carbon dioxide (CO_2_) and an atmospheric half-life of 8.6 years [5] versus 50–200 years for CO_2_ [6]. Therefore, reducing the production of enteric CH_4_ provides a practical avenue to rapidly reduce the environmental impact of livestock systems.

Many approaches to mitigate enteric CH_4_ production have been evaluated for their efficacy [4], and dietary supplementation with CH_4_-inhibiting feed additives has proven most effective to date. However, many of the most potent inhibitors (e.g., red seaweeds, bromochloromethane, chloroform) come with considerable caveats, including the risk of adverse side effects damaging to the environment, animals, and/or downstream consumers, and may provide only transient reductions in CH_4_ [7]. A synthetic CH_4_ inhibitor, 3-nitrooxypropanol (3-NOP, marketed as Bovaer™, DSM Nutritional Products, Basel, Switzerland), has recently been approved in numerous countries for the reduction in enteric CH_4_ from ruminant livestock, and has no known toxicity risks [8]. 3-NOP acts by inhibiting methyl coenzyme-M reductase, the enzyme that catalyzes the terminal step of archaeal methanogenesis [9]. Multiple studies have shown that its addition to basal diets decreases enteric CH_4_ emissions by 20–80% in beef and dairy cattle, though effectiveness varies according to dietary composition and dose [10,11,12,13].

Lipid supplementation is another potent and cost-effective strategy for reducing enteric CH_4_ emissions. Dietary lipids can lower CH_4_ production by up to 30%, though their effectiveness depends on the lipid source, dose, and basal diet composition [14,15,16]. Additionally, lipid inclusion can alter the fatty acid profile of milk and meat, enhancing their health benefits for human consumers [17,18]. The mechanism of lipid-induced CH_4_ reduction differs amongst lipid sources and includes toxicity toward protozoa and methanogens, promotion of biohydrogenation as an alternative sink for metabolic H_2_, and redirection in rumen fermentation to favor propionate synthesis (an alternative H_2_ sink), thus reducing net H_2_ production [7,19]. Lipids also reduce the proportion of fermentable organic matter in the diet, which also contributes to reduced CH_4_ [19]. However, high proportions of dietary lipids reduce fiber degradation and dry matter intake via the inhibition of fiber-degrading bacteria and protozoa, causing significant changes in the rumen bacterial community [20,21]. These undesirable side effects must therefore be balanced with the CH_4_-mitigating properties of dietary lipid supplementation. Combining two or more CH_4_ inhibitors simultaneously can increase the potency of CH_4_ inhibition and can balance potentially negative side effects of a single inhibitor, but few studies have been conducted to date [13,22]. In two previously published companion studies, we found that co-supplementation with 3-NOP and canola oil more effectively suppressed enteric CH_4_ emissions than either inhibitor alone [13,20]. This effect was accompanied by increased propionate production, reduced rumen fiber digestibility, and alterations in the rumen bacterial, archaeal, and protozoal communities [13,20].

The effects of CH_4_ inhibitors on rumen bacterial, archaeal, and protozoal communities are well documented, but there is limited knowledge of the response of the rumen fungi to standalone or combined CH_4_ inhibition strategies. The anaerobic gut fungi (AGF) of the rumen belong to the phylum Neocallimastigomycota [23,24] and account for up to 20% of the rumen microbial biomass [25]. They play a critical role in microbial degradation of plant cell wall carbohydrates, as demonstrated by the decrease in feed intake and dry matter digestion that followed their removal from the rumen of sheep [26]. Rumen fungi display a syntrophic relationship with the methanogenic archaea by facilitating interspecies H_2_ transfer during enteric methanogenesis. This interaction between rumen fungi and methanogens has also been shown to enhance the fiber-degrading capacity of the fungi by upregulating the expression of multiple fibrolytic enzymes [27,28,29]. It is unclear whether disrupting the close interaction between methanogens and AGF by selective inhibition of rumen methanogens also alters the fungal community in the short or longer term. Given the key role that anaerobic fungi play in feed digestibility efficiency, it is essential to understand whether feed additives negatively impact these microbes and whether this impact persists beyond the withdrawal of treatment.

The impact of 3-NOP and canola oil, fed alone and in combination, on the rumen bacteria and archaea has been investigated [11,20], but there is currently no information available concerning their effect on the rumen fungi. The present study has addressed this shortfall by applying amplicon sequencing of the D1/D2 domain of the fungal LSU gene to samples of ruminal content collected from animals offered 3-NOP, canola oil, or both additives in combination. Our findings point to a deleterious effect of high concentrations of canola oil on the rumen AGF, which at least in part explains lipid-induced depressions in ruminal fiber digestion previously reported [13,16].

## 2. Materials and Methods

### 2.1. Experimental Design and Dietary Treatments

The animal trial was conducted at the Agriculture and Agri-Food Canada Research and Development Centre in Lethbridge, AL, Canada. A comprehensive description of the materials and methods has been previously published [13,20]. In brief, eight ruminally cannulated Angus crossbred beef heifers (732 ± 43 kg) were arranged in a replicated 4 × 4 Latin square consisting of four 28-day periods and four dietary treatments arranged in a 2 × 2 factorial design (Table 1): 3-NOP (with or without) and canola oil (with or without). Each period included a 13-day adaptation phase followed by a 15-day measurement and sample collection phase. The dietary treatments were as follows: (1) control—basal diet; (2) 3-NOP—supplementation with 3-NOP alone (Bovaer^TM^ fed at 200 mg/kg diet dry matter [DM], DSM Nutritional Products Ltd., Kaiseraugst, Switzerland); (3) oil—canola oil supplementation alone (50 g/kg DM, oil; Loveland Industries, Inc., Loveland, CO, USA); and (4) 3-NOP + oil—a combination of 3-NOP (Bovaer fed at 200 mg/kg DM) and canola oil (50 g/kg DM). The animals were offered a high-forage total mixed ration (TMR) comprising 900 g/kg barley silage, 41.2 g/kg dry-rolled barley grain, 50 g/kg supplement mix, and 8.8 g/kg treatment mix (control or treatment; DM basis). Treatment mixes were prepared weekly, refrigerated, and fully incorporated into the diet before feeding. TMRs were prepared daily and fed at 10:00 h. On day 14 of each period, rumen content samples were collected at three time points: pre-feeding (0 h) and 6 and 12 h post-feeding. A representative 1 L sample of ruminal content (both solid and liquid phases) was obtained by collecting and combining digesta from the ventral, caudal, and dorsal–ventral sacs within the rumen, as well as from the reticulum. The collected rumen content samples were filtered through two layers of polyester monofilament fabric (355 µm mesh) to separate the solid and liquid fractions. Filtered rumen fluid and digesta samples were snap-frozen in liquid nitrogen and stored at −80 °C until DNA extraction.

### 2.2. Metagenomic DNA Extraction

Frozen rumen samples were freeze-dried and ground using a coffee grinder, which was sanitized thoroughly between samples. Microbial DNA was extracted from 0.1 g of the ground material using the Zymobiomics DNA extraction kit per the manufacturer’s instructions (Zymo Research, Irvine, CA, USA). Concentration and purity of the extracted metagenomic DNA were determined by measuring ratios of absorbance at 260/280 nm and 260/230 nm using a NanoDrop spectrophotometer (Thermo Fisher Scientific, Mississauga, ON, Canada). Extracted DNA was stored at −80 °C pending molecular analysis.

### 2.3. Quantitative Real-Time Polymerase Chain Reaction (qPCR)

All qPCR reactions were performed on the QuantStudio 6 Real-Time PCR system (Applied biosystems; Thermo Fisher Scientific) using absolute quantification settings and PerfeCTa^®^ SYBR^®^ Green SuperMix, Low ROX™ (Quantabio, Beverly, MA, USA). The primers LSU-F (5′-GCGTTTRRCACCASTGTTGTT-3′) and LSU-R (5′-GTCAACATCCTAAGYGTAGGTA-3′) were used to amplify the D1/D2 region of the fungal 28S-LSU gene [30]. Purified PCR products were subcloned into the pCR™2.1 Vector at a 1:3 ratio (plasmid:insert) using the Invitrogen™ TA Cloning™ Kit and transformed into One Shot™ TOP10 Chemically Competent *E. coli* cells (Invitrogen; Thermo Fisher Scientific) following the manufacturer’s instructions. Plasmids were isolated with QIAprep Spin Miniprep Kit (Qiagen) and confirmed to contain the 355 bp insert using Sanger sequencing (Genome Quebec). The standard curve consisted of 7 points of a tenfold dilution series of a known concentration of plasmid, and a no-template control (NTC) was included in all experiments. Each standard, sample, and NTC was run in triplicate in a total reaction volume of 10 μL. Reactions contained 5 μL 2× MasterMix, gene specific primers (500 nM), 1 μL of the diluted (1:10) DNA template and/or plasmid and DNase/RNase free water to 10 µL The reactions were performed at 95 °C for 15 min, 40 cycles of 95 °C for 15 s, 60 °C for 45 s, and 72 °C for 30 s. The specificity of the PCR products was verified by melting curve analysis after each run by increasing the temperature at a rate of 1 °C every 60 s from 60 to 95 °C. The slope, amplification efficiency (E%), and coefficient of determination (R^2^), along with quantification cycle (C_q_) values and Dgene copy numbers of each sample, were calculated using the QuantStudio Real-Time PCR software version 1.7.2 (Applied biosystems; Thermo Fisher Scientific, Mississauga, ON, Canada). The concentration of fungal LSU was converted to gene copy numbers using the QuantStudio Real-Time PCR software version 1.7.2. Outliers were identified and removed if the Coefficient of variation (CV) was >17%, but no samples contained fewer than 2 data points. Final copy numbers were normalized to reflect starting DNA concentration per gram of dry rumen sample.

### 2.4. Amplicon Sequencing

The D1/D2 hypervariable domain of the LSU (28S rRNA gene) was amplified using the same primers used for qPCR ligated to Illumina adapters (Illumina, San Diego, CA, USA) [30] under the following conditions: denaturation for 30 s at 98 °C followed by 33 cycles of 98 °C for 10 s, 60 °C for 30 s, 72 °C for 30 s, and a final extension of 72 °C for 2 min. A 5 µL aliquot from each reaction was used to verify amplification in a 1% agarose gel with 1× TAE buffer and 1× GelRed™ Nucleic Acid Gel Stain (Gold Biotechnology; Cedarlane, Burlington, ON, Canada). The samples were then shipped on dry ice to Genome Quebec Innovation Centre (Montreal, QC, Canada) for all subsequent wet lab analysis and sequencing. Samples were quantified using the Quant-iT PicoGreen dsDNA Assay Kit (Life Technologies, Carlsbad, CA, USA) and were pooled in equal proportions. Pooled samples were then purified using calibrated Ampure XP beads (Beckman Coulter, Mississauga, ON, Canada). The pooled samples (library) were quantified using the Quant-iT PicoGreen dsDNA Assay Kit (Life Technologies, Carlsbad, CA, USA) and the Kapa Illumina GA with Revised Primers-SYBR Fast Universal kit (Kapa Biosystems, Wilmington, MA, USA). Average fragment size was determined using a LabChip GX (PerkinElmer, Waltham, MA, USA) instrument. Negative extraction and PCR controls, as well as a positive control with amplicons from genomic DNA extracted from known pure AGF cultures, were included alongside experimental samples. Amplicon sequencing was performed using an Illumina NextSeq platform (Illumina, San Diego, CA, USA) following the manufacturer’s guidelines.

### 2.5. Bioinformatic and Statistical Analyses

Raw fastq-format files were imported into Qiime2-2023.5 for sequence analysis [31]. Primer and adapter sequences were removed from sequence files using the Cutadapt plugin (v.5.0) [32]. Sequence quality score plots were manually evaluated, and both forward and reverse primers were truncated at position 250 bp. Sequences were denoised into amplicon sequencing variants (ASVs) using DADA2 (v.1.16) [33], and chimeras were simultaneously removed. Multiple sequence alignment and masking of highly variable regions were performed using MAFFT (version 7) [34]. A phylogenetic tree was generated using FastTree (v.2.1) [35]. The ASVs were taxonomically classified using a Naïve Bayes classifier trained on the curated collection of D1/D2-LSU reference sequences provided by the Anaerobic Fungi Network (https://anaerobicfungi.org/). ASV count matrix and phylogenetic tree files were imported to R (v.4.4.2) [36] using the ‘qiime2R’ package (v.0.99) [37]. Samples with fewer than 100 sequences in total following denoising were removed. ASV and genus prevalence were computed using custom R programming. Count data were normalized using log-transformation (with a pseudocount addition of 1) prior to α- and β-diversity calculation. α-Diversity, measured using the Shannon Index [38], was calculated with Phyloseq (v.1.5) [39]. Statistically significant differences between groups were identified using an ANOVA (on log-normalized data) with Tukey’s post-hoc tests. β-diversity was visualized using Principal Coordinate Analysis (PCoA) plots based on Generalized Unifrac distances [40]. Redundancy analyses (RDA) were performed using Vegan (v.2.6) [41] and MicroViz (v.0.12) [42]. PERMANOVA and β-dispersion tests were performed using Vegan. Spearman correlations were performed using base R code alongside the corrplot [43] package. All figures were prepared using the ggplot2 package (v.3.5.1) [44]. Differential abundance testing between control and all other groups was performed at the genus level using a linear model implemented in ANCOM-BC2 [45]. Real-time PCR data were normalized and compared between treatments using a linear mixed effects model implemented in the lmerTest R package [46] with Tukey’s post hoc tests. For all tests, statistically significant differences were declared at *p* < 0.05, with trends at *p* < 0.1. False discovery rate (FDR) correction for multiple testing was applied where necessary, and *q* < 0.05 was used to indicate statistically significant differences, with trends at *q* < 0.1.

The ability of the AGF to recover to their baseline state following dietary intervention (i.e., their resilience) was measured at community and individual taxonomic levels. AGF profiles in samples collected from animals who received the control diet in period 1 (P1-CON) were used as the baseline dataset and compared with the AGF profiles of samples collected from control-fed animals during periods 2 (P2-CON), 3 (P3-CON), and 4 (P4-CON) (See Table 1 for period–treatment combinations). PERMANOVA and Procrustes tests (computed using Vegan) were applied to measure resilience at community level. To measure recovery at the taxonomic level, a resilience index (i.e., a measure of the ability of a given genus to return to its baseline control sample abundance) was calculated for each genus in the dataset using the following formula:(1)$${RES}_{i,j}=\frac{A_{i,j}}{A_{i,1}}$$
where ${RES}_{i,j}$ is the resilience index for genus i during period j (where j ≠ 1), $A_{i,j}$ is the abundance of genus i during period j, and $A_{i,1}$ is the abundance of genus i during the baseline period (j = 1).

The abundance of each genus in P1-CON animals was used as the baseline. The baseline abundance of each genus was then compared with the abundance of each genus from the P2-CON, P3-CON, and P4-CON samples. From these comparisons, three measures of resilience (RES_P2-CON_, RES_P3-CON_, and RES_P4-CON_) were calculated for each genus, corresponding to the specific control feeding period following the inhibitor treatments. A RES value of 1 in P2, P3, or P4 indicates that the genus has returned to its baseline abundance in that period; a value greater than 1 signifies that its abundance exceeds the baseline, whereas a value less than 1 shows that it has not recovered to the baseline abundance.

## 3. Results

### 3.1. Sequencing Characteristics

The 160 samples sequenced successfully contained on average 17,012 ± 5234 classified reads, ranging from 125 to 32,562 per sample. The reads were distributed among 201 ASVs assigned to 18 genera and a single phylum, Neocallimastimycota. Nine genera were present in more than 10% of the samples (Figure 1A). All rarefaction curves reached a plateau between 5000 and 10,000 sequences (Figure 1B), indicating that sequencing was conducted to a sufficient depth.

### 3.2. Effect of Methane Inhibitors on Fungal Diversity and Community Structure

α-Diversity, measured by the Shannon Index, was significantly altered by treatment and period (*p* < 0.05). Shannon diversity was reduced by oil (*p* < 0.05) and the combination of 3NOP + oil (*p* < 0.05) compared with the control samples (Figure 1C). PERMANOVA tests showed strong and significant effects of treatment (R^2^ = 0.36, *p* < 0.05) and the interaction of treatment and period (R^2^ = 0.16; *p* < 0.05) on fungal community composition. Ordination plots using the Generalized Unifrac distances confirmed the similarity of control and 3NOP samples, while oil and 3NOP + oil samples were distinct and more dispersed (*p* < 0.05, Figure 1D). β-dispersion analysis showed significantly higher intra-group variability for the oil and 3NOP + oil samples compared with the control and 3NOP samples (*p* < 0.05, Figure 1E), which likely contributed partially to the high R^2^ value attributed to the treatment effect in the PERMANOVA test. Detailed test results are provided in the Appendix A. When summarized at the genus level, *Neocallimastix, Caecomyces*, and *Piromyces* were predominant (Figure 1F).

Constrained ordination analysis (RDA) confirmed the distinct differences in microbial community composition across the treatment groups seen in the PCoA. The RDA plot showed a strong relationship between the control and 3NOP samples and higher proportions of all the major VFAs, as well as H_2_ (Figure 2A). Methane concentration exhibited an intermediate relationship between the 3NOP/control cluster and the oil/3NOP + oil cluster. Higher abundance of most of the fungal genera was more strongly associated with the 3NOP and control samples, except for *Liebetanzomyces* and *Caecomyces,* which exhibited a stronger relationship with the oil and 3NOP + oil samples.

### 3.3. Effect of Methane Inhibitors on Abundance of Fungal Genera

Prior to differential abundance testing, ASVs that could not be assigned to a genus were combined as “unknown” taxa, but as these may, in theory, represent multiple undescribed genera and species with divergent functions, we have not inferred any biological meaning from the associated tests. Results are provided in Table 2 and Figure 2B. The control samples were used as the reference level for each comparison. When fed alone, 3NOP increased the abundance of NY08 and reduced that of *Liebetanzomyces* (*p* < 0.05). The abundances of NY08 and *Caecomyces* (trend, *p* < 0.1) were enriched by oil treatment, while those of *Piromyces*, *Neocallimastix*, NY05, and NY08 were reduced (*p* < 0.05). The combination of 3NOP + oil induced statistically significant changes in the abundances of each genus compared with controls, with NY08, *Caecomyces*, and *Liebetanzomyces* all enriched and *Neocallimastix, Piromyces,* NY05, and NY09 all suppressed (*p* < 0.05).

### 3.4. Community and Individual Genus Resilience to CH_4_ Inhibitors

The ability of the AGF to return to their baseline composition was assessed using only the samples collected from control-fed animals during each period. PERMANOVA tests showed there were significant differences in AGF composition between each group in this reduced dataset (*p* < 0.05, R^2^ = 0.41). Pairwise tests showed that P1-CON samples were significantly different from P2-CON, P3-CON, and P4-CON samples (*p* < 0.05; Figure 3A). This was supported by Procrustes tests (*p* > 0.05), which provided the same results. α-Diversity, measured by the Shannon index, was lower in P2-CON and P4-CON compared with both P1-CON and P3-CON (*p* < 0.05; Figure 3B). Detailed test results are provided in the Appendix A.

To assess the resilience of individual taxa, a resilience index was calculated for the genera present in more than 10% of the samples. *Caecomyces* was the only genus to surpass its baseline (P1-CON) abundance in each subsequent control-fed period (RES_P2-CON_ = 1.09, RES_P3-CON_ = 1.06, RES_P4-CON_ = 1.02; Table 3, Figure 3C). In contrast, *Neocallimastix* did not fully recover its abundance at any subsequent period (RES_P2-CON_ = 0.75, RES_P3-CON_ = 0.82, RES_P4-CON_ = 0.77). Several other genera exhibited an initial increase in abundance following perturbation before declining below their baseline during period 4 (e.g., Piromyces, RES_P2-CON_ = 1.28, RES_P3-CON_ = 1.10, RES_P4-CON_ = 0.88).

### 3.5. Quantitative Real-Time PCR Results

Confirming trends seen in the sequencing data, control and 3NOP samples had greater quantities of LSU genes than both oil and 3NOP + oil treatment groups (*p* < 0.05), but there were no differences between CON and 3NOP, or between oil and 3NOP + oil groups (Figure 4A). There was also a significant interaction between treatment and period (*p* < 0.05). When the qPCR expression data were plotted by period and treatment, it was apparent that the deleterious effect of oil-containing treatments on LSU gene quantity was less when supplemented during periods 3 and 4 than during periods 1 and 2 (Figure 4B). A comprehensive statistical analysis of qPCR data is provided in the Appendix A.

## 4. Discussion

Rapidly lowering the carbon footprint of dairy and beef production will require the development of effective strategies for reducing enteric CH_4_ emissions. We previously reported the effects of 3-nitrooxypropanol (3-NOP, Bovaer™), canola oil, and their combination on enteric CH_4_ emissions, rumen fermentation, and the microbial ecology of the rumen bacterial, archaeal, and protozoal communities [13,20]. To briefly summarize these findings, 3-NOP alone reduced CH_4_ yield (g/kg DMI) by 31.6%, while canola oil alone reduced it by 27.4%, consistent with previous studies [7]. Combining the feed supplements resulted in an additive CH_4_ yield reduction of 51.4% compared with control animals. Diets containing 3-NOP induced targeted modifications in rumen bacterial and archaeal communities, redirected rumen fermentation towards alternative H_2_ sinks, and enhanced propionate production. In contrast, canola oil supplementation reduced methanogen abundance, caused widespread shifts in rumen bacterial composition, eliminated rumen protozoa and key fibrolytic bacteria, and altered VFA concentrations and proportions [20]. Notably, CH_4_-mitigating levels of canola oil also reduced fiber digestibility by 20% [13].

Despite the critical role AGF play in the rumen microbial ecosystem and fiber digestion, the impact of CH_4_-mitigating additives on the AGF remains underexplored. This knowledge gap is largely due to the difficulty in isolating and maintaining AGF in pure culture, as well as the lack of consensus on an appropriate molecular marker gene [28,47]. Sequencing of the 18S rRNA and internal transcribed spacer (ITS1/2) genes has been widely used in metabarcoding studies of fungal communities for some time [48,49,50,51]. However, several limitations constrain the utility of these markers for studies targeting the Neocallimastigomycota. The 18S gene lacks the sequence variability required to reliably differentiate among AGF taxa. Additionally, while the ITS1 region has been widely used as a universal marker within the fungal kingdom [52], it has been found to be polymorphic within the AGF, displaying considerable variability in both length and sequence even within a single fungal strain [53,54,55,56]. Differences in specificity between primer sets have also led to underrepresentation of the true AGF (i.e., the Neocallimastigomycota), with disproportionate amplification of aerobic fungi, including molds and yeasts with no known role in rumen function [48]. Exploration of alternatives has identified the D1/D2 region of the 28S rRNA subunit (LSU) as a promising alternative to the ITS and 18S genes, with a more conserved size and sequence across species [53,57]. Additionally, the recent publication of a comprehensive set of reference sequences for the D1/D2 region provides an excellent platform for robust metabarcoding studies of the ruminal AGF [23].

To our knowledge, this study is the first to use amplicon sequencing of the LSU gene to examine the relationship between AGF and CH_4_ mitigation strategies. Overall, the data show that despite its significant effects on CH_4_ production, 3-NOP fed alone had minimal impact on the rumen AGF. In contrast, canola oil supplementation—both alone and in combination with 3-NOP—caused substantial disruption to the structure and diversity of the AGF community in the rumen. Interestingly, individual AGF genera exhibited contrasting responses to lipid supplementation, likely due to differences in their resilience to environmental stressors. Additionally, we found that repeated perturbations of the ruminal AGF via dietary manipulation impair their ability to return to baseline composition, which may contribute to the altered fiber degradation and rumen function that was observed previously [13].

### 4.1. Canola-Oil-Containing Diets Cause Substantial Changes in AGF Community Composition, Diversity, and LSU Gene Quantity

Lipid supplementation can be an effective strategy for reducing enteric CH_4_ emissions [13], but the high concentrations at which they must be included decrease fiber and feed digestibility [15,16]. These effects are primarily attributed to lipid toxicity toward cellulolytic bacteria and ciliate protozoa, as well as reduced feed digestibility when lipids replace dietary carbohydrate sources [19,20]. Early studies suggested that lipid supplementation also inhibits the AGF [58,59], but to our knowledge, this is the first study to employ direct sequencing of the ruminal AGF to measure their response to dietary lipids in vivo. Our data reveal that canola oil exerts a pronounced antifungal effect in the rumen, with notable differences in response among AGF taxa. Specifically, its supplementation—either alone or in combination with 3-NOP—led to significant reductions in fungal diversity, overall LSU gene concentrations, and in the abundance of key genera such as *Neocalimastix* and *Piromyces,* as well as uncultured genera including NY05 and NY09. In contrast, the abundance of *Caecomyces* and the uncultured genus NY08 was significantly increased.

The mechanisms underlying these differential responses cannot be fully understood based on the available data but likely stem from the unique biochemical and morphological traits of different AGF species [27,47]. Canola oil is rich in unsaturated fatty acids (UFAs), including linolenic, linoleic, and oleic acids [60]. Microbial lipolysis in the rumen releases UFAs, which can disrupt microbial membranes when present in their free form, leading to cellular lysis and death [61]. This toxicity can be partially counteracted by rumen microbes via biohydrogenation, but this process has limited capacity [62,63]. Variations in fungal growth patterns and morphology may also contribute to these differences. *Caecomyces* exhibits a bulbous rhizoidal growth pattern, whereas *Neocalimastix* and *Piromyces* grow in a filamentous pattern [64]. A recent global census of ruminant AGF diversity found that *Caecomyces* is more common in the lower gut than in the rumen, and proposed that their bulbous growth and unique attachment mechanisms may enhance survival during gastrointestinal transit by protecting the fungal thallus within the plant biomass [65]. A similar protective effect may explain its resistance to lipid toxicity, but further research is needed to confirm this hypothesis.

AGF are known to digest substantially more fiber than rumen bacteria [66]. The earliest described and most intensively studied of the AGF, *Neocallimastix* spp. [47,65] are highly effective degraders of a range of recalcitrant lignocellulosic substrates including cellulose and xylan [28,67]. In contrast, *Caecomyces* exhibit a preference for short-chain polysaccharides and monosaccharides, including glucose and fructose [68]. The suppression of *Neocalimastix* and *Piromyces* by canola oil likely contributed to the reduction in fiber digestibility reported in our companion study [13]. While AGF produce the VFA, acetate, they do not synthesize propionate or butyrate [69], so the suppression of AGF metabolism in the rumen may contribute to the reduced acetate concentration observed in the canola-oil-supplemented cattle [13]. Ruminal degradation of plant biomass is a synergistic process involving bacteria, fungi, and protozoa [70], and our previous work has shown that the addition of dietary canola oil, with or without 3-NOP, significantly reduces the abundance of major ciliates, including *Isotricha*, *Dasytricha*, and *Entodinium,* as well as keystone cellulolytic bacteria like *Fibrobacter* spp. [20]. Our findings here show that lipid supplementation affects all major microbial groups in the rumen, likely contributing to the additive inhibition of ruminal fiber digestion and fermentation, generally as a side effect of enteric methanogenesis reduction.

Our findings underscore that while canola oil supplemented with or without 3-NOP is effective in reducing enteric CH_4_, it significantly alters the ruminal AGF in a manner that almost certainly contributes to reduced fiber digestion and VFA production. Differences in sensitivity to canola oil among different AGF may be linked to differences in fungal morphology and growth dynamics. Future studies should focus on understanding the basis of this differential sensitivity, not only in AGF but in ruminal bacteria and protozoa as well. Such insights will be critical in developing dietary strategies that can balance CH_4_ mitigation with stable ruminal fermentation patterns.

### 4.2. 3-NOP Mitigates Enteric Methanogenesis Without Impacting the Rumen Anaerobic Fungi

In contrast to canola oil, supplementation with 3-NOP alone did not significantly alter the composition or diversity of ruminal AF, despite resulting in a ~32% reduction in CH_4_ yield [13]. Previous studies have shown that 3-NOP does not alter the digestibility or microbial colonization of feed in the rumen and has mixed effects on the rumen bacterial and archaeal communities [11,20,71,72,73]. The overall composition of the bacterial community remains largely stable; however, the metabolic shifts and changes in H_2_ levels associated with 3-NOP supplementation elicit targeted bacterial responses. These include reduced expression in H_2_-producing bacteria [73], stimulation of rumen acetogens [74], and alterations in the abundance of *Methanobrevibacter*, *Methanosphaera, Ruminococcus, Butyrivibrio* [73], and *Candidatus Faecousia* [74]. The data presented here allow us to extend this understanding of how 3-NOP impacts the rumen microbiome to AGF and show that it has no substantial impact on AGF composition, diversity, or LSU gene concentration. The targeted nature of 3-NOP is a direct consequence of its highly selective mode of action, acting as a competitive inhibitor of the methyl-coenzyme M reductase (MCR) enzyme, which is ubiquitously and uniquely found in methanogenic archaea [10,75]. This specificity allows 3-NOP to effectively inhibit CH_4_ production without substantially impacting other microbial groups in the rumen. There was a minor response to 3-NOP supplementation among AGF genera, with a decrease in the abundance of *Liebetanzomyces* and an increase in that of the uncharacterized genus NY08. NY08 was identified as a novel fungal genus in a comprehensive census of the herbivorous mycobiome [76], but has not yet been formally described or named, so it is difficult to conclude what implications the increase in its abundance may have for rumen function. Similarly, little is known of the precise functional role of *Liebetanzomyces* in the rumen. It produces acetate in vitro but has reduced β-glucosidase activity versus other major anaerobic fungi, indicating that it does not play a prominent role in the digestion of cellulose [77]. The fact that only minor taxa respond to 3-NOP indicates that the concurrent alterations in rumen fermentation patterns arise from niche-specific changes in less dominant AGF genera, rather than from broad-scale shifts in the overall mycobiome [13].

Methanogens and AGF exist in an ectosymbiotic relationship, which provides H_2_ for archaeal methanogenesis while stimulating AGF hydrolytic enzyme secretion [27,78,79]. Our findings here and previously [13,20] reveal that despite the significant impact of 3-NOP on enteric methanogenesis and the overall rumen archaeal population, feed digestibility is unaffected, and the AGF community remains stable. This raises an intriguing question: if the AGF–archaea interaction enhances lignocellulose digestion, why does its disruption not lead to a decline in feed utilization? One possibility is that while free-living archaea are inhibited by 3-NOP, those in direct contact with AGF may receive some level of protection, thereby preserving the functional benefits of their interaction. Alternatively, it is possible that in a complex and functionally redundant ecosystem like the rumen, other hydrogenotrophs or even host-mediated processes partially compensate for the absence of the methanogen-provided services that support AGF metabolism. Given the essential role of AGF in fiber degradation and overall feed efficiency in ruminants [28,69], these results support the potential of targeted CH_4_ mitigation strategies that selectively inhibit methanogenesis without compromising key microbial functions in the rumen. Moreover, our study suggests that AGF metabolism remains robust even in the presence of elevated hydrogen concentrations associated with 3-NOP supplementation [10], further underscoring the feasibility of such approaches.

### 4.3. Repeated Perturbation of the Rumen Fungi with Canola Oil Reduces the Ability of Certain AGF Taxa to Return to Their Pre-Supplementation State

Resilience is defined as the ability of a community to resist or rapidly recover from disturbances [80], and is crucial for efficient rumen microbiome function. Our study shows that repeated exposure to CH_4_ inhibitors induces lasting changes in the rumen mycobiome, potentially compromising AGF resilience under continuous perturbation. Analysis of control-fed samples across four periods (P1-CON [baseline], P2-CON, P3-CON, and P4-CON) revealed that repeated challenges alter AGF composition, even after reverting to a normal diet. PERMANOVA and Procrustes tests revealed significant shifts in community structure over time in this reduced dataset, with pairwise comparisons among samples from control-fed animals showing significant differences between all groups. Interestingly, fungal α-diversity in the P3-CON samples returned to baseline levels. This is likely attributable to the sequence of treatments—animals in P3-CON received oil in P1, 3-NOP in P2, and the control diet in P3 (Figure 4A). As discussed above, 3-NOP alone did not substantially alter the AGF, and this extended period of dietary stability allowed AGF diversity to recover. However, PCoA showed that even with this recovery in diversity, the P3-CON AGF community was still compositionally distinct. This result suggests a differential ability of different AGF species to recover following perturbation, based on their level of tolerance to environmental stress.

To explore this further, we calculated a resilience index for individual genera. In this index, values exceeding 1 indicate recovery beyond baseline levels, whereas values below 1 reflect incomplete recovery. This analysis confirmed our hypothesis, with clear differences in resilience among AGF genera. *Caecomyces* consistently exhibited RES values greater than 1, suggesting a robust capacity to recover following CH_4_ inhibitor exposure. In contrast, several genera, including *Anaeromyces*, NY07, and NY15, were not detected in certain periods, indicating a higher sensitivity to the perturbations and that they might be more transient members of the rumen AGF. The proliferation of *Caecomyces* in later periods contrasted with the consistently low RES values of *Neocalimastix* and *Piromyces* throughout the experiment, indicative of their lower resilience to perturbation. As discussed above, *Neocalimastix* and *Piromyces* are robust fiber degraders, while *Caecomyces* spp. show a preference for simpler polymers and monomers [28,81]. Thus, the recovery of AGF diversity does not necessarily point to recovery in fiber degradation capacity if fewer fibrolytic AGF are becoming predominant. This could be more comprehensively studied using metagenomic or metatranscriptomic sequencing to quantify the expression of microbial enzymes responsible for fiber digestion. Additionally, long-term studies are needed to determine whether the observed compositional changes are reversible over extended periods or if they represent lasting alterations to the rumen AGF. Finally, our findings also indicate that Latin square designs may not be appropriate for evaluating the effect of CH_4_ inhibitors on the rumen microbiome, or at least that a longer washout period between treatments should be used.

## 5. Conclusions

This study demonstrates that while both canola oil and 3-NOP are effective in reducing enteric CH_4_ emissions, their impacts on the rumen AGF are markedly different. Canola oil, whether administered alone or in combination with 3-NOP, significantly disrupts AGF community structure by reducing overall diversity, LSU gene concentrations, and the abundance of key fiber-degrading taxa such as *Neocallimastix* and *Piromyces*. In contrast, 3-NOP supplemented alone achieves substantial CH_4_ mitigation without negatively affecting the AGF community, highlighting its potential as a targeted CH_4_ abatement strategy that preserves essential rumen functions. Finally, our data suggest that repeated perturbation may have a persistent and potentially negative impact on the AGF, and that the effect of CH_4_ inhibitors should be studied over a longer period of time to elucidate the long-term implications of these microbial shifts on rumen efficiency and overall animal productivity.

## Figures and Tables

**Figure 1 animals-15-01230-f001:**
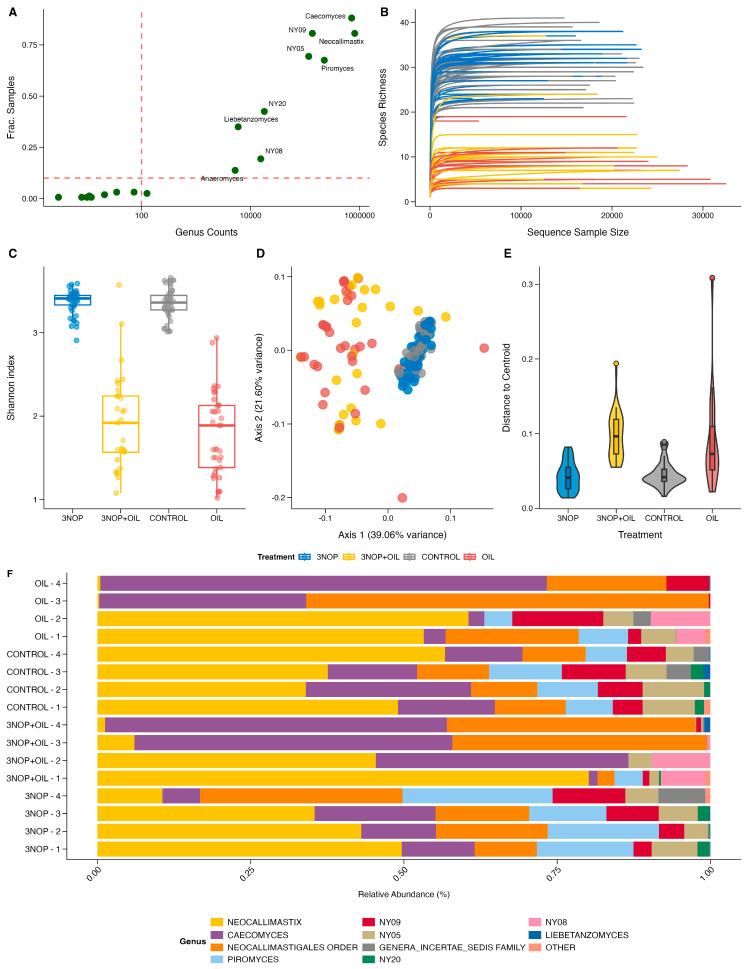
Overview of microbial community structure and diversity across experimental treatments and samples (N = 160): (**A**) Genera prevalence plot showing the number of samples in which each genus was detected. (**B**) Rarefaction curves (step = 100) illustrating sequencing depth and sample richness. (**C**) Boxplots of alpha diversity indices by treatment group. (**D**) Principal coordinates analysis (PCoA) based on a Generalized UniFrac distance. (**E**) Violin plots depicting beta dispersion across treatment groups. (**F**) Stacked bar chart of genus-level taxonomic composition, grouped by treatment and sampling period. The 10 most abundant taxa overall are presented, with the remainder folded into “Other”.

**Figure 2 animals-15-01230-f002:**
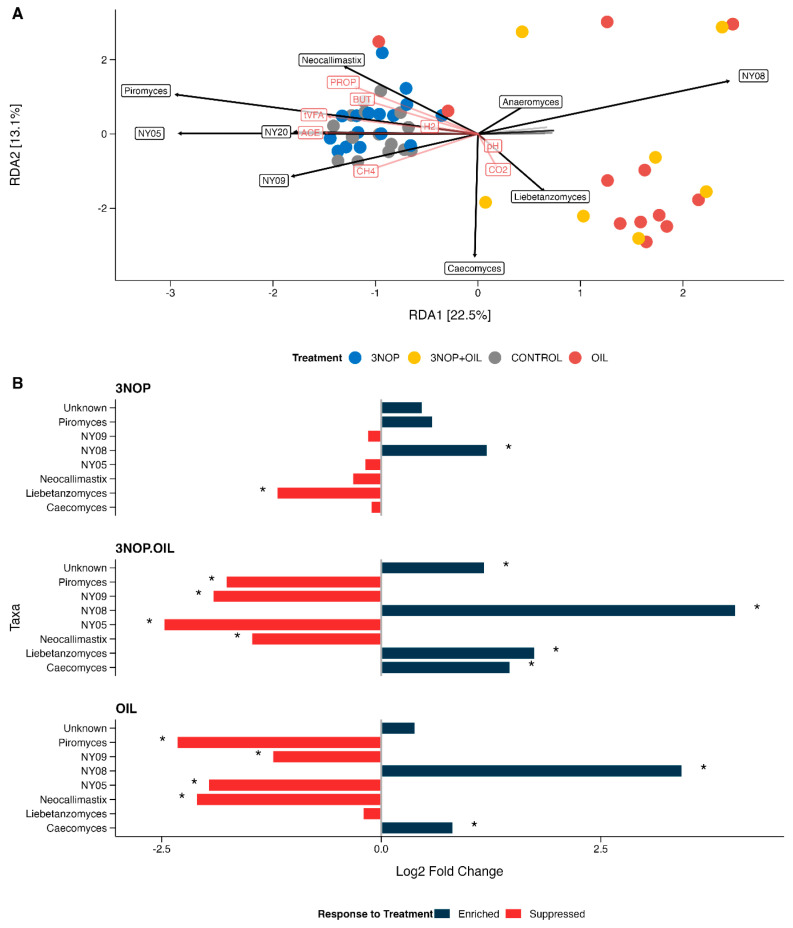
(**A**) Redundancy analysis (RDA) ordination plots of microbial communities under different treatments (3-NOP, 3-NOP + oil, control, and oil). Colored points represent individual samples, and arrows indicate the direction and relative magnitude of taxa associated with each RDA axis. (**B**) Differential abundance analysis of major taxa across treatment groups using ANCOM-BC2. Bars extending to the right (positive values) indicate enrichment, whereas bars to the left (negative values) indicate suppression relative to the control animals. Asterisks denote statistically significant differences and trends (*p* < 0.05, *p* < 0.1).

**Figure 3 animals-15-01230-f003:**
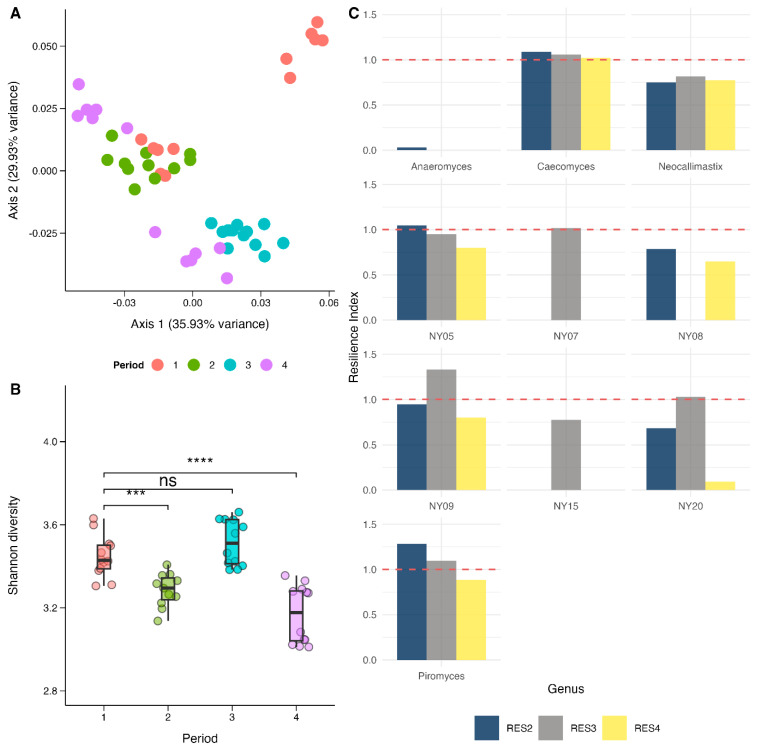
Microbial resilience analysis plots, using only the control samples from each period: (**A**) Principal coordinates analysis (PCoA) of the microbial community using a Generalized UniFrac distance, showing distinct clustering by sampling period (colors). (**B**) Boxplots of Shannon diversity across four sampling periods. Statistical comparisons are indicated by brackets, with “ns” denoting no significant difference, “***” indicating *p <* 0.001, and “****” indicating *p* < 0.0001. (**C**) Resilience Index values for key anaerobic fungal genera. The dashed red line represents the baseline reference threshold for resilience.

**Figure 4 animals-15-01230-f004:**
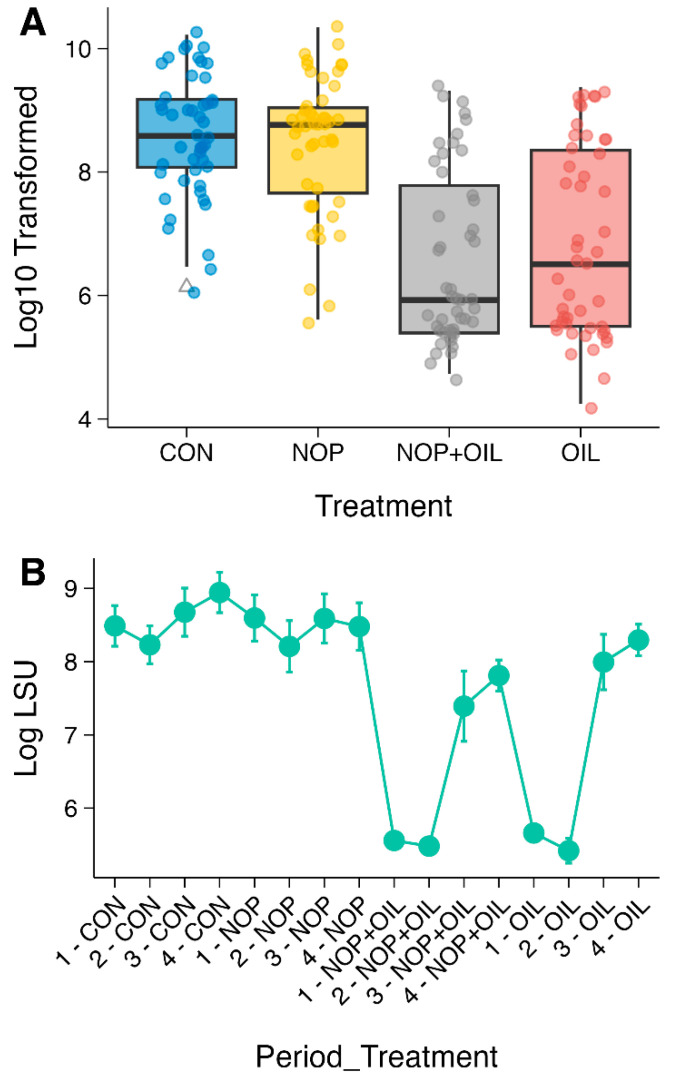
(**A**) Boxplots showing the log_10_-transformed total LSU expression across four treatment groups (CON, 3-NOP, 3-NOP + oil, oil). Each point represents an individual sample. (**B**) Line plot of normalized LSU expression (log scale) by sampling period and treatment combination, illustrating temporal variation in LSU levels under each experimental condition.

**Table 1 animals-15-01230-t001:** Treatment order in the 4 × 4 Latin Square Design employed in this experiment. Control-fed periods are highlighted for clarity.

Period	11, 21 *	9, 10	6, 12	2, 8
P1	** CONTROL **	3-NOP + OIL	OIL	3-NOP
P2	OIL	** CONTROL **	3-NOP	3-NOP + OIL
P3	3-NOP + OIL	3-NOP	** CONTROL **	OIL
P4	3-NOP	OIL	3-NOP + OIL	** CONTROL **

* Denotes the animal numbers allocated to each Treatment-Period combination.

**Table 2 animals-15-01230-t002:** Differential Abundance Analysis using ANCOM-BC2. Log_2_ fold changes (Log2FC) and corresponding adjusted *p*-values (P-adj.) for comparisons between the control treatment (control) and three experimental treatments (oil, 3-NOP, and 3-NOP + oil) are presented.

	CONTROL vs. OIL		CONTROL vs. 3-NOP		CONTROL vs. 3-NOP + OIL	
Genus	Log2FC	P-adj.	Log2FC	P-adj.	Log2FC	P-adj.
*Liebetanzomyces*	−0.21	0.61	−1.19	0.02	1.75	<0.01
*Caecomyces*	0.82	0.08	−0.12	0.75	1.47	<0.01
*NY05*	−1.97	<0.01	−0.19	0.71	−2.48	<0.01
*NY08*	3.43	<0.01	1.21	0.02	4.04	<0.01
*NY09*	−1.24	<0.01	−0.16	0.71	−1.92	<0.01
*Neocallimastix*	−2.11	<0.01	−0.33	0.54	−1.48	<0.01
*Piromyces*	−2.33	<0.01	0.59	0.27	−1.77	<0.01
Unknown taxa	0.39	0.36	0.47	0.35	1.18	<0.01

**Table 3 animals-15-01230-t003:** Genus-level resilience values relative to the baseline (P1-CON) for all subsequent control-fed periods. Higher values indicate greater resilience relative to the control.

Genus	RES_P2-CON_	RES_P3-CON_	RES_P4-CON_
*Piromyces*	1.28	1.10	0.88
*Caecomyces*	1.09	1.06	1.02
*NY05*	1.05	0.95	0.80
*NY09*	0.95	1.33	0.80
*NY08*	0.79	0.00	0.65
*Neocallimastix*	0.75	0.82	0.77
*NY20*	0.68	1.03	0.09
*Anaeromyces*	0.03	ND *	ND
*NY07*	ND	1.02	ND
*NY15*	ND	0.77	ND

***** ND = not detected.

## Data Availability

Raw sequences have been deposited in the Short Reads Archive (SRA) under accession number PRJNA1202537.

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
