# Peer review of "Assessing the Impact of the Methane Inhibitors 3-Nitrooxypropanol (3-NOP) and Canola Oil on the Rumen Anaerobic Fungi"

_animals, 2025, doi:10.3390/ani15091230_

Round 1
Reviewer 1 Report
Comments and Suggestions for Authors
In the manuscript “Assessing the impact of the methane inhibitors 3-nitrooxypropanol (3-NOP) and canola oil on the rumen anaerobic fungi” the authors investigated the effect of dietary 3-nitrooxypropanol (3-NOP) and canola oil supplementation on the community of Neocallimastigomycota (AGF) in fistulated cattle. The experimental set-up and data analysis is clearly laid out and easy to follow. The introduction provides relevant background information on the topic and explains the necessity for the study well. The conclusion puts the results of this study into context of the general literature as well as the companion study on the bacterial community. Overall, the manuscript is of high-quality, presents novelty, and, I believe, will be of interest to the scientific community. There are, however, some issues in the presentation of the results, which should be taken care of before publication.
Issues that should be taken care of:
Figure 1F:
- Could you group the identified genera here in the same way you do later-on in the manuscript? (i.e., the seven main groups and the rest as “Others”). Also, the classification level should be unified to genus level. I assume the family and order level groups in this plot represent AGF that could not be classified any further, but I would suggest to group those together as “Unclassified AGF” (or such) anyways. High-level taxonomic delineations within Neocallimastigomycota are currently unresolved due to lack of data and – as you correctly state in your manuscript – uncultured genera are unfortunately not useful for the interpretation of your results because we don’t know anything about them yet. You could provide the (unclustered) details of your results in a supplementary OTU count table.
- Related issue: Could you maybe provide a supplementary table detailing the number of sequences you’ve obtained for each sample (after all quality control steps, etc)? This information could be another column in the above-mentioned supplementary table. The range is quite high (125-32,562) and it might be interesting to see which samples (I assume OIL and the mixed treatment) are on the lower end and which are on the higher end.
- The explanation what “Other” are, is missing in the figure legend. Are those genera with relative abundance <0.01%, <0.001%?
Figure 2B, Table 2, and the text in lines 286-297:
- The information given in the figure, in the table and in the text seems to be contradictory, e.g., in the text NY08 and Caecomyces are stated to be reduced in OIL treatment compared to control while the figure and the table show them to be enriched. The reverse is true for the other genera in the OIL treatment samples. Additionally, Figure 3B shows the Log2 Fold Change in Caecomyces OIL treatment to be significant, while the text and the table do not.
- Also, it might be good to pick a P-value limit (<0.01 or <0.05) and use that consistently throughout the manuscript (in the text, figures, and tables) and the supplement when reporting significance limits instead of exact numbers.
- Given that table 2 and Figure 2B are redundant, I would suggest to move the table to the supplement, and keep the text and Figure 2B in the main manuscript.
Figure 3A:
- I am having a hard time understanding what the PCoA in Figure3A is based on. The figure legend states that only the CONTROL samples from each period are used, but if that is the case shouldn’t there only be 2 points per color (since there were only 2 samples per CONTROL) in each period? If the PCoA plot is based on the resilience calculations described in the method section, shouldn’t all P1 CONTROL values be the same (1) and hence cluster together? I assume that the PCoA is based on the overall beta diversity found in the samples, but it might be good to state that more clearly in the figure legend.
- There seems to be a pattern of clusters within P1 and P4 that is not talked about in the result section and – as described above – I am missing the details about these clusters (they are neither the figure, nor in the manuscript or the supplementary) to figure out what those clusters could be. Could you maybe provide more clarity and information on the PCoA and maybe touch on that clustering pattern either in the results or in the discussion section of the manuscript somewhere?
Suggestions/minor things:
Line 23-24: I think there might be a word missing (“rumen anaerobic community/structure (?)”)
Line 97: I would suggest removing the word “hypervariable” here to avoid confusion. This region is moderately variable in AGF, making it conserved enough to be specifically targeted in environmental samples, but variable enough amongst AGF to differentiate them into species.
Line 143-44 and line 168-169: You are using the same primer pair under 2 different names (LSU-F/AGF-LSU-EnvS-F and LSU-R/AGF-LSU-EnvS-R)
Line 311-312: You state here that resilience index was only calculated for genera present in more than 10% of samples. If you are referring to the 8 control samples, that would mean each singe sample. If you are referring to all samples you have, I am wondering why not more rare genera are included in addition to Anaeromyces?
Lines 359-380: The information given here could be shorted a little bit and moved into the introduction, since it is more background information than discussion/contextualization of the results.
Line 403-404: instead of “are of unknown function” you could write “uncultured” because even though those AGF are uncultured we can reasonably assume that they have broadly similar function (fiber degradation)
Line 421: reference citation formatting
Line 422: instead of “most abundant and widespread AGF” it might be better to refer to it as the “first described and one of the most intensely studied AGF” (or something like that) given that Neocallimastix is not the most abundant AGF (even though it is widespread; but so is Piromyces, for example).
Line 427: There’s a stray comma here.
Line 427 and Lines 469-471: You state in 427 that AGF do not produce propionate or butyrate (which is not entirely true, at least as far as I know), and in line 469-471 you talk about the production of propionate and butyrate in Liebetanzomyces.
Line 482f: I would like to see one alternative explanation mentioned here (e.g., that in complex environments maybe the service provided to AGF by methanogens can be fulfilled by other organisms or even the host itself), given that the exact nature and depth of the symbioses between AGF and methanogens is still not clear (it could still be a simple matter of extracellular nutrient exchange or inhibitor removal).
The references are not yet consistently formatted.
Author Response
In the manuscript “Assessing the impact of the methane inhibitors 3-nitrooxypropanol (3-NOP) and canola oil on the rumen anaerobic fungi” the authors investigated the effect of dietary 3-nitrooxypropanol (3-NOP) and canola oil supplementation on the community of Neocallimastigomycota (AGF) in fistulated cattle. The experimental set-up and data analysis is clearly laid out and easy to follow. The introduction provides relevant background information on the topic and explains the necessity for the study well. The conclusion puts the results of this study into context of the general literature as well as the companion study on the bacterial community. Overall, the manuscript is of high-quality, presents novelty, and, I believe, will be of interest to the scientific community. There are, however, some issues in the presentation of the results, which should be taken care of before publication.
Response from Authors: Thank you for taking the time to provide a comprehensive review of our manuscript.
Issues that should be taken care of:
Figure 1F:
- Could you group the identified genera here in the same way you do later-on in the manuscript? (i.e., the seven main groups and the rest as “Others”). Also, the classification level should be unified to genus level. I assume the family and order level groups in this plot represent AGF that could not be classified any further, but I would suggest to group those together as “Unclassified AGF” (or such) anyways. High-level taxonomic delineations within Neocallimastigomycota are currently unresolved due to lack of data and – as you correctly state in your manuscript – uncultured genera are unfortunately not useful for the interpretation of your results because we don’t know anything about them yet. You could provide the (unclustered) details of your results in a supplementary OTU count table.
Response from Authors: Thank you for your thoughtful feedback regarding Figure 1F. We appreciate your suggestion to group the identified genera into the seven main groups and classify the remainder as “Others,” as well as your recommendation to unify the taxonomic classification level to genus. However, we respectfully maintain our current approach for the following reasons.
While we acknowledge that certain ASVs could not be classified beyond the family or order level, they have nonetheless been identified as taxonomically distinct based on alignment with a curated reference database. These features likely represent uncultured or uncharacterized genera within Neocallimastigomycota, and although high-level taxonomic delineations within this phylum remain unresolved, combining these taxa into a single “Unclassified AGF” category would obscure potentially meaningful differences. The primary aim of our study is to evaluate the effect of dietary treatments (3-NOP, canola oil) on the rumen anaerobic fungal community. Given the possibility that these currently unclassified taxa respond differently to treatment and/or time (period) - even responding in opposite ways - aggregating them could result in the loss of important ecological signals.
While we agree that uncultured taxa currently offer limited functional interpretation, we believe their inclusion provides a more accurate representation of the diversity and dynamics of the anaerobic fungal community across treatments and periods in this study. As cultivation efforts expand, the taxonomy of many of these currently uncultured groups may become better resolved. Thus, we view our findings as contributing foundational data that future studies may build upon.
To ensure transparency and facilitate interpretation, and also in response to your next comment, we have provided the complete taxonomic table in the Supplementary Material (Table S1), which includes all classified features and their abundances. We hope this addresses your concern while clarifying our rationale for retaining the current taxonomic resolution in Figure 1F.
- Related issue: Could you maybe provide a supplementary table detailing the number of sequences you’ve obtained for each sample (after all quality control steps, etc)? This information could be another column in the above-mentioned supplementary table. The range is quite high (125-32,562) and it might be interesting to see which samples (I assume OIL and the mixed treatment) are on the lower end and which are on the higher end.
Response from Authors: Thank you for your suggestion. We agree that this is useful information to include, and have added it to the Supplementary Data.
- The explanation what “Other” are, is missing in the figure legend. Are those genera with relative abundance <0.01%, <0.001%?
Response from Authors: Thank you for pointing out this oversight. This Figure presents the 10 most abundant genera, and the remainder are folded into “Other”. This information has been added to the legend for Figure 1. It is also worth noting that relative proportions are not used for any testing or comparisons in this paper, and this figure is to give the reader a general idea of microbial composition across periods and treatments.
Figure 2B, Table 2, and the text in lines 286-297:
- The information given in the figure, in the table and in the text seems to be contradictory, e.g., in the text NY08 and Caecomyces are stated to be reduced in OIL treatment compared to control while the figure and the table show them to be enriched. The reverse is true for the other genera in the OIL treatment samples. Additionally, Figure 3B shows the Log2 Fold Change in Caecomyces OIL treatment to be significant, while the text and the table do not.
Response from Authors: Thank you for your careful reading. We have confirmed that the figure contains the correct data and have revised the text on Lines 291-292 to align with the figure and table. Additionally, we clarified in the text that the “*” symbol in Figure 2B is used to denote both statistical significance and trends, which may have contributed to confusion.
- Also, it might be good to pick a P-value limit (<0.01 or <0.05) and use that consistently throughout the manuscript (in the text, figures, and tables) and the supplement when reporting significance limits instead of exact numbers.
Response from Authors: Thank you for this helpful suggestion. We have reviewed the manuscript and ensured consistent use of P-value thresholds where appropriate, while retaining exact values in some contexts to provide clarity and transparency for readers.
- Given that table 2 and Figure 2B are redundant, I would suggest to move the table to the supplement, and keep the text and Figure 2B in the main manuscript.
Response from Authors: We appreciate this suggestion. While Figure 2B provides a visual summary, we believe that retaining Table 2 in the main manuscript adds value by providing detailed numerical context for genus-level changes. Given that the manuscript includes only two tables, we feel Table 2 is not excessive and supports reader interpretation of the results.
Figure 3A:
- I am having a hard time understanding what the PCoA in Figure3A is based on. The figure legend states that only the CONTROL samples from each period are used, but if that is the case shouldn’t there only be 2 points per color (since there were only 2 samples per CONTROL) in each period? If the PCoA plot is based on the resilience calculations described in the method section, shouldn’t all P1 CONTROL values be the same (1) and hence cluster together? I assume that the PCoA is based on the overall beta diversity found in the samples, but it might be good to state that more clearly in the figure legend.
Response from Authors: Thank you for this comment. This has been stated very explicitly from lines 216-220.
As you will see in the Methods section (L119-126), we collected three samples on the day of sampling ( 0 Hour, 3H, 6H). Each of these samples was fractioned into the solid and liquid phase, and processed separately. This gave 6 samples per animal. Not every one of these samples was successfully sequenced, so there are a few less and certain periods. In our initial analysis we tested for differences according to Hour and sample fraction, finding that there was no discernable effect. Therefore in order to increase the power of our analysis, we treated these samples as technical replicates within each animal-period combination, while controlling for them appropriately in all statistical modelling.
Regarding the second part of this comment; we agree it would be optimal to see all P1-CON samples cluster nicely together, but unfortunately as we know only too well, the microbiome is a highly variable beast! There are definitely sub-clusters evident within some of the Periods, which are an unfortunate side effect of using a latin square design. In our study, we employed a replicated 4 x 4 design, meaning essentially that 2 Groups of 4 animals underwent the same experiment. Each square/group was staggered by a week to accommodate the availability of the open-circuit respiratory chambers (n=4). This factor was controlled for statistically whenever required. This also addresses your below comment.
- There seems to be a pattern of clusters within P1 and P4 that is not talked about in the result section and – as described above – I am missing the details about these clusters (they are neither the figure, nor in the manuscript or the supplementary) to figure out what those clusters could be. Could you maybe provide more clarity and information on the PCoA and maybe touch on that clustering pattern either in the results or in the discussion section of the manuscript somewhere?
Response from Authors: See above.
Suggestions/minor things:
Line 23-24: I think there might be a word missing (“rumen anaerobic community/structure (?)”)
Response from Authors: Thank you for pointing out this oversight, this has been addressed in the revised manuscript.
Line 97: I would suggest removing the word “hypervariable” here to avoid confusion. This region is moderately variable in AGF, making it conserved enough to be specifically targeted in environmental samples, but variable enough amongst AGF to differentiate them into species.
Response from Authors: Thank you for the suggestion and clear contextualization. We have removed the word ‘hypervariable’ to avoid confusing the reader.
Line 143-44 and line 168-169: You are using the same primer pair under 2 different names (LSU-F/AGF-LSU-EnvS-F and LSU-R/AGF-LSU-EnvS-R)
Response from Authors: Thank you for pointing this out. These are indeed the same amplification primers, and the divergence in nomenclature comes from our in-house labelling system to distinguish between qPCR and sequencing primers (which have Illumina adapters ligated to them). We have amended the text (L167) to avoid confusion.
Line 311-312: You state here that resilience index was only calculated for genera present in more than 10% of samples. If you are referring to the 8 control samples, that would mean each singe sample. If you are referring to all samples you have, I am wondering why not more rare genera are included in addition to Anaeromyces?
Response from Authors: Thank you for pointing this out. We believe we have addressed this in our below responses, as this derived from a lack of clarity on the number of samples used for this portion of the analysis.
Lines 359-380: The information given here could be shorted a little bit and moved into the introduction, since it is more background information than discussion/contextualization of the results.
Response from Authors: Thank you for your suggestion. We have scrutinized this section, and appreciate your perspective. However, in this case we believe that the information presented in this section is highly relevant to readers who are involved in anaerobic gut fungi research, particularly in the rumen. The lack of a reliable marker gene for the rumen AF has been a consistent issue for researchers, and we believe that this detailed context will provide valuable information for all stakeholders in the field, beyond an abridged version that would be included in the introduction.
Line 403-404: instead of “are of unknown function” you could write “uncultured” because even though those AGF are uncultured we can reasonably assume that they have broadly similar function (fiber degradation)
Response from Authors: thank you for your suggestion, we have amended the text as suggested.
Line 421: reference citation formatting
Response from Authors: Thank you for pointing out this oversight, we have corrected the error.
Aiken et al
Line 422: instead of “most abundant and widespread AGF” it might be better to refer to it as the “first described and one of the most intensely studied AGF” (or something like that) given that Neocallimastix is not the most abundant AGF (even though it is widespread; but so is Piromyces, for example).
Response from Authors: Thank you for your comment. We have reworded this sentence for clarity and conciseness.
Line 427: There’s a stray comma here.
Response from Authors: Thank you for pointing this out, this has been amended in the revised manuscript.
Line 427 and Lines 469-471: You state in 427 that AGF do not produce propionate or butyrate (which is not entirely true, at least as far as I know), and in line 469-471 you talk about the production of propionate and butyrate in Liebetanzomyces.
Response from Authors: Thank you for your comment. We have revised the literature and believe we are correct in our statement that the true anaerobic fungi of the rumen do not produce propionate or butyrate. This is stated / shown in:
- Peng et al., 2021. DOI : https://doi.org/10.1038/s41564-020-00861-0
- Edwards et al., DOI: https://doi.org/10.3389/fmicb.2017.01657
- Gordon & Phillips, 1998. DOI: https://doi.org/10.1079/NRR19980009
In particular, the last review provides citations for several pioneer studies that demonstrated AGF produce only acetate of the major VFA.
In this context, we acknowledge that our phrasing on lines 469-471 was confusing to the reader. We have amended this for clarity on L466-469 of the revised manuscript.
Line 482f: I would like to see one alternative explanation mentioned here (e.g., that in complex environments maybe the service provided to AGF by methanogens can be fulfilled by other organisms or even the host itself), given that the exact nature and depth of the symbioses between AGF and methanogens is still not clear (it could still be a simple matter of extracellular nutrient exchange or inhibitor removal).
Response from Authors: Thank you for your suggestion, we have added some further discussion to this section on lines 482-485 of the revised manuscript.
The references are not yet consistently formatted.
Response from Authors: Thank you for this observation, we have striven to ensure all references are now correctly formatted.
Reviewer 2 Report
Comments and Suggestions for Authors
Strength: This study utilizes novel sequencing tools for gut fungi, hence serving as a technical guidance in exploring the gut mycobiome in general. The paper is well written, neatly organized. The paper is written at a level understandable for general audience, except for the methodology jargons. The figures are clear and well presented. The authors did a great job explaining the data, integrating different components of the study and tying back to the original question.
Weakness: The resilience index study is based on a small sample size and therefore the power of that analysis is questionable. A baseline sampling of all the animals in the trial at the beginning would have been better.
Page 2, line 60: “lipid inclusion can alter the fatty acid profile of milk and meat enhancing their health benefits for human consumers” – this statement is confusing. Does inclusion of lipids in the diet increase or decrease fat content of milk and meat?
Page 2, lines 61-64: The sentence is complicated, please re-write this. What does "alternative sink for metabolic H2" mean?
Page 3, line 107: Are the beef heifers age-matched?
Page 3, line 110: What measurements and samples were collected during the 15 days? Later it says the rumen content sampling was done on the 14th day. How many samples from each animal were used for sequencing analyses?
Page 5, lines 217-221: it seems that data used in resilience measurement has very small sample size (n=2). Please justify this small sample size and the analysis used.
Page 6, line 242: Please explain how you have 160 samples? Does this include replicates? or different time points within each period?
Page 7, lines 268-269 (Fig 1 legend): Please indicate sample size for these analyses
Page 8, line 270: Is this clustered ordination or redundancy analysis. In the figure legend it is indicated as redundancy analysis
Page 9, lines 312-315: NY05 (period 2) and NY09 (period 3) also surpassed their baseline values
Page 15, lines 497-499: This conclusion is based on a small sample size of 2 (P1-CON) vs each of the other periods (n=2 for each); please explain and justify your sample size
Author Response
Strength: This study utilizes novel sequencing tools for gut fungi, hence serving as a technical guidance in exploring the gut mycobiome in general. The paper is well written, neatly organized. The paper is written at a level understandable for general audience, except for the methodology jargons. The figures are clear and well presented. The authors did a great job explaining the data, integrating different components of the study and tying back to the original question.
Response from Authors: Thank you for your comprehensive and constructive review of our manuscript.
Weakness: The resilience index study is based on a small sample size and therefore the power of that analysis is questionable. A baseline sampling of all the animals in the trial at the beginning would have been better.
Response from Authors: Thank you for your comment regarding the resilience index analysis. On day 14, rumen contents were sampled at three time points (0 h, 3 h, and 6 h), and each was separated into solid and liquid fractions - giving six samples per animal. Not all samples were successfully sequenced, leading to slight variation in sample numbers across treatments and time points. Initial analyses indicated no significant effect of sampling hour or fraction, so we treated these as technical replicates and combined them to increase statistical power. These were controlled for statistically where necessary. We hope that this also addresses your below comment about the sampling period.
We agree that having baseline (pre-treatment) samples from all animals would have strengthened the study design and allowed for a more robust assessment of individual microbial trajectories. However, this unfortunately was not performed at the start of the trial due to budgetary and logistical constraints. Despite this limitation, we believe the analysis still provides meaningful insights into microbial stability and treatment effects over time within the available dataset.
Page 2, line 60: “lipid inclusion can alter the fatty acid profile of milk and meat enhancing their health benefits for human consumers” – this statement is confusing. Does inclusion of lipids in the diet increase or decrease fat content of milk and meat?
Response from Authors: Thank you for your comment. Our intent was not to suggest that lipid inclusion necessarily increases the total fat content of milk or meat, but rather that certain dietary lipids can alter the composition of the fat - specifically, by increasing the proportion of beneficial mono- and poly-unsaturated fatty acids. This change in fatty acid profile is generally regarded as an enhancement of the nutritional quality of animal products for human consumption.
Page 2, lines 61-64: The sentence is complicated, please re-write this. What does "alternative sink for metabolic H2" mean?
Response from Authors: Thank you for highlighting this. This is the common terminology used in the field to describe the flow of the hydrogen that is produced during the metabolic process taking place in the rumen. High levels of environmental hydrogen inhibit microbial metabolism, and so inhibit fermentation in the rumen. Methane is the primary hydrogen sink, so inhibiting this process results in opportunities to redirect the flow of hydrogen into alternative sources like the production of the VFA propionate. We have added more explicit detail on L63 of the revised manuscript.
Page 3, line 107: Are the beef heifers age-matched?
Response from Authors: The heifers were of comparable age and were blocked by body weight. Mean body weight (BW ± SD) was 732 ± 43 kg
Page 3, line 110: What measurements and samples were collected during the 15 days? Later it says the rumen content sampling was done on the 14th day. How many samples from each animal were used for sequencing analyses?
Response from Authors: Thank you for your question. During the 15-day experimental period, we collected multiple measurements, including body weight, feed intake, methane emissions, and feed/fecal samples, as detailed in our companion manuscripts. On day 14, rumen contents were sampled as described in our above response.
Page 5, lines 217-221: it seems that data used in resilience measurement has very small sample size (n=2). Please justify this small sample size and the analysis used.
Response from Authors: Thank you for your comment. This is related to our previous responses on sample numbers. Only the control-fed samples from each period were used, but there were up to 6 samples per animal-period combination, so this vastly increased the power of this study. We hope that these additional details and the previous responses to reviewer comments have provided clarity.
Page 6, line 242: Please explain how you have 160 samples? Does this include replicates? or different time points within each period?
Response from Authors: Thank you for your question. The methods indicate the time points that samples were collected, as well as the number of animals included in the experiment. Each sample was split into rumen liquid and rumen solid and analyzed separately. In total metagenomic DNA was extracted from 196 samples however due to the disruptive nature of the OIL treatment a number of the samples from these did not result in amplification or sequences for AGF. This left 160 samples that hade sequence data that could be analyzed.
Page 7, lines 268-269 (Fig 1 legend): Please indicate sample size for these analyses
Response from Authors: Thank you for your suggestion. This was based on all 160 samples. We have included an explicit statement in the figure legend indicating that the total number of samples included in the analysis that were used to generate figure 1 is N=160.
Page 8, line 270: Is this clustered ordination or redundancy analysis. In the figure legend it is indicated as redundancy analysis
Response from Authors. Thank you for your question. This is redundancy analysis, which is a extension of regression, in that it models a response matrix onto an explanatory matrix. It is constrained analysis by it’s nature.
Page 9, lines 312-315: NY05 (period 2) and NY09 (period 3) also surpassed their baseline values
Response from Authors: We thank the reviewer for their comment. We agree with this observation. We believe we have captured these and similar trends on line 315-316, stating “Several other genera exhibited an initial increase in abundance following perturbation before declining below their baseline during period 4 (e.g. Piromyces, RESP2-CON = 1.28, RESP3-CON = 1.10, RESP4-CON = 0.88).”
Page 15, lines 497-499: This conclusion is based on a small sample size of 2 (P1-CON) vs each of the other periods (n=2 for each); please explain and justify your sample size
Response from Authors: Thank you for your comment. The responses regarding the calculation of resilience index have addressed this. Due to the lack of statistical difference between the samples taken at times 0,6,12 hour in the liquid and solid phases we treated these samples as technical replicates which increased the number of samples used in this analysis.